# Neural Language Modeling
# by Jointly Learning Syntax and Lexicon

**Yikang Shen, Zhouhan Lin, Chin-Wei Huang & Aaron Courville**
Department of Computer Science and Operations Research
Universit de Montral
Montral, QC H3C3J7, Canada
`{yi-kang.shen, zhouhan.lin, chin-wei.huang, aaron.courville}@umontreal.ca`

## Abstract

We propose a neural language model capable of unsupervised syntactic structure induction. The model leverages the structure information to form better semantic representations and better language modeling. Standard recurrent neural networks are limited by their structure and fail to efficiently use syntactic information. On the other hand, tree-structured recursive networks usually require additional structural supervision at the cost of human expert annotation. In this paper, We propose a novel neural language model, called the Parsing-Reading-Predict Networks (PRPN), that can simultaneously induce the syntactic structure from unannotated sentences and leverage the inferred structure to learn a better language model. In our model, the gradient can be directly back-propagated from the language model loss into the neural parsing network. Experiments show that the proposed model can discover the underlying syntactic structure and achieve state-of-the-art performance on word/character-level language model tasks.

## 1 Introduction

Linguistic theories generally regard natural language as consisting of two part: a *lexicon*, the complete set of all possible words in a language; and a *syntax*, the set of rules, principles, and processes that govern the structure of sentences (Sandra & Taft, 1994). To generate a proper sentence, tokens are put together with a specific syntactic structure. Understanding a sentence also requires lexical information to provide meanings, and syntactical knowledge to correctly combine meanings. Current neural language models can provide meaningful word represent (Bengio et al., 2003; Mikolov et al., 2013; Chen et al., 2013). However, standard recurrent neural networks only implicitly model syntax, thus fail to efficiently use structure information (Tai et al., 2015).

Developing a deep neural network that can leverage syntactic knowledge to form a better semantic representation has received a great deal of attention in recent years (Socher et al., 2013; Tai et al., 2015; Chung et al., 2016). Integrating syntactic structure into a language model is important for different reasons: 1) to obtain a hierarchical representation with increasing levels of abstraction, which is a key feature of deep neural networks and of the human brain (Bengio et al., 2009; LeCun et al., 2015; Schmidhuber, 2015); 2) to capture complex linguistic phenomena, like long-term dependency problem (Tai et al., 2015) and the compositional effects (Socher et al., 2013); 3) to provide shortcut for gradient back-propagation (Chung et al., 2016).

A syntactic parser is the most common source for structure information. Supervised parsers can achieve very high performance on well constructed sentences. Hence, parsers can provide accurate information about how to compose word semantics into sentence semantics (Socher et al., 2013), or how to generate the next word given previous words (Wu et al., 2017). However, only major languages have treebank data for training parsers, and it request expensive human expert annotation. People also tend to break language rules in many circumstances (such as writing a tweet). These defects limit the generalization capability of supervised parsers.

Unsupervised syntactic structure induction has been among the longstanding challenges of computational linguistic (Klein & Manning, 2002; 2004; Bod, 2006). Researchers are interested in this

problem for a variety of reasons: to be able to parse languages for which no annotated treebanks exist (Marecek, 2016); to create a dependency structure to better suit a particular NLP application (Wu et al., 2017); to empirically argue for or against the poverty of the stimulus (Clark, 2001; Chomsky, 2014); and to examine cognitive issues in language learning (Solan et al., 2003).

In this paper, we propose a novel neural language model: Parsing-Reading-Predict Networks (PRPN), which can simultaneously induce the syntactic structure from unannotated sentences and leverage the inferred structure to form a better language model. With our model, we assume that language can be naturally represented as a tree-structured graph. The model is composed of three parts:

1. **A differentiable neural Parsing Network** uses a convolutional neural network to compute the *syntactic distance*, which represents the syntactic relationships between all successive pairs of words in a sentence, and then makes soft constituent decisions based on the syntactic distance.

2. **A Reading Network** that recurrently computes an adaptive memory representation to summarize information relevant to the current time step, based on all previous memories that are syntactically and directly related to the current token.

3. **A Predict Network** that predicts the next token based on all memories that are syntactically and directly related to the next token.

We evaluate our model on three tasks: word-level language modeling, character-level language modeling, and unsupervised constituency parsing. The proposed model achieves (or is close to) the state-of-the-art on both word-level and character-level language modeling. The model's unsupervised parsing outperforms some strong baseline models, demonstrating that the structure found by our model is similar to the intrinsic structure provided by human experts.

## 2 RELATED WORK

The idea of introducing some structures, especially trees, into language understanding to help a downstream task has been explored in various ways. For example, Socher et al. (2013); Tai et al. (2015) learn a bottom-up encoder, taking as an input a parse tree supplied from an external parser. There are models that are able to *infer* a tree during test time, while still need supervised signal on tree structure during training. For example, (Socher et al., 2010; Alvarez-Melis & Jaakkola, 2016; Zhou et al., 2017; Zhang et al., 2015), etc. Moreover, Williams et al. (2017) did an in-depth analysis of recursive models that are able to learn tree structure without being exposed to any grammar trees. Our model is also able to infer tree structure in an unsupervised setting, but different from theirs, it is a recurrent network that implicitly models tree structure through attention.

Apart from the approach of using recursive networks to capture structures, there is another line of research which try to learn recurrent features at multiple scales, which can be dated back to 1990s (e.g. El Hihi & Bengio (1996); Schmidhuber (1991); Lin et al. (1998)). The NARX RNN (Lin et al., 1998) is another example which used a feed forward net taking different inputs with predefined time delays to model long-term dependencies. More recently, Koutnik et al. (2014) also used multiple layers of recurrent networks with different pre-defined updating frequencies. Instead, our model tries to learn the structure from data, rather than predefining it. In that respect, Chung et al. (2016) relates to our model since it proposes a hierarchical multi-scale structure with binary gates controlling intra-layer connections, and the gating mechanism is learned from data too. The difference is that their gating mechanism controls the updates of higher layers directly, while ours control it softly through an attention mechanism.

In terms of language modeling, syntactic language modeling can be dated back to Chelba (1997). Charniak (2001); Roark (2001) have also proposed language models with a top-down parsing mechanism. Recently Dyer et al. (2016); Kuncoro et al. (2016) have introduced neural networks into this space. It learns both a discriminative and a generative model with top-down parsing, trained with a supervision signal from parsed sentences in the corpus. There are also dependency-based approaches using neural networks, including Buys & Blunsom (2015); Emami & Jelinek (2005); Titov & Henderson (2010).

Parsers are also related to our work since they are all inferring grammatical tree structure given a sentence. For example, SPINN (Bowman et al., 2016) is a shift-reduce parser that uses an LSTM as its composition function. The transition classifier in SPINN is supervisedly trained on the Stanford PCFG Parser (Klein & Manning, 2003) output. Unsupervised parsers are more aligned with what our model is doing. Klein & Manning (2004) presented a generative model for the unsupervised learning of dependency structures. Klein & Manning (2002) is a generative distributional model for the unsupervised induction of natural language syntax which explicitly models constituent yields and contexts. We compare our parsing quality with the aforementioned two papers in Section 6.3.

## 3 MOTIVATION

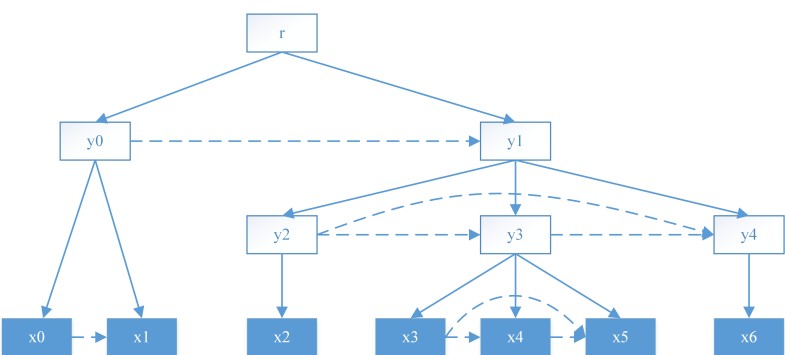

Figure 1: Hard arrow represents syntactic tree structure and parent-to-child dependency relation, dash arrow represents dependency relation between siblings

Suppose we have a sequence of tokens $x_0, ..., x_6$ governed by the tree structure showed in Figure 1. The leafs $x_i$ are observed tokens. Node $y_i$ represents the meaning of the constituent formed by its leaves $x_{l(y_i)}, ..., x_{r(y_i)}$, where $l(\cdot)$ and $r(\cdot)$ stands for the leftmost child and right most child. Root $r$ represents the meaning of the whole sequence. Arrows represent the dependency relations between nodes. The underlying assumption is that each node depends only on its parent and its left siblings.

Directly modeling the tree structure is a challenging task, usually requiring supervision to learn (Tai et al., 2015). In addition, relying on tree structures can result in a model that is not sufficiently robust to face ungrammatical sentences (Hashemi & Hwa, 2016). In contrast, recurrent models provide a convenient way to model sequential data, with the current hidden state only depends on the last hidden state. This makes models more robust when facing nonconforming sequential data, but it suffers from neglecting the real dependency relation that dominates the structure of natural language sentences.

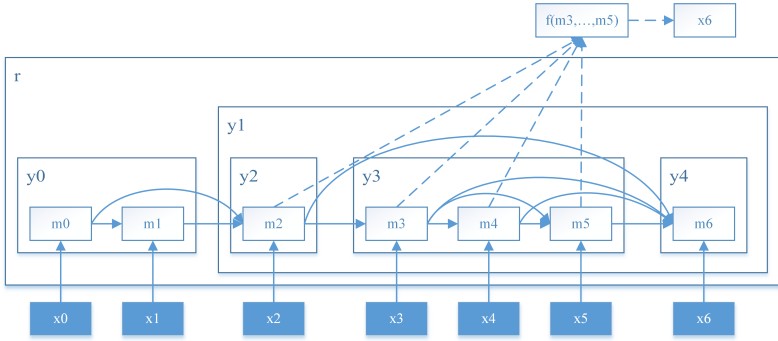

Figure 2: Proposed model architecture, hard line indicate valid connection in Reading Network, dash line indicate valid connection in Predict Network.

In this paper, we use skip-connection to integrate structured dependency relations with recurrent neural network. In other words, the current hidden state does not only depend on the last hidden state, but also on previous hidden states that have a direct syntactic relation to the current one.

Figure 2 shows the structure of our model. The non-leaf node $y_j$ is represented by a set of hidden states $y_j = \{m_i\}_{l(y_j) \leq i \leq r(y_j)}$, where $l(y_j)$ is the left most descendant leaf and $r(y_j)$ is the right most one. Arrows shows skip connections built by our model according to the latent structure. Skip connections are controlled by gates $g_i^t$. In order to define $g_i^t$, we introduce a latent variable $l_t$ to represent local structural context of $x_t$:

- if $x_t$ is not left most child of any subtree, then $l_t$ is the position of $x_t$'s left most sibling.
- if $x_t$ is the left most child of a subtree $y_i$, then $l_t$ is the position of the left most child that belongs to the left most sibling of $y_i$.

and gates are defined as:

$$g_i^t = \begin{cases} 1, & l_t \leq i < t \\ 0, & 0 < i < l_t \end{cases} \tag{1}$$

Given this architecture, the siblings dependency relation is modeled by at least one skip-connect. The skip connection will directly feed information forward, and pass gradient backward. The parent-to-child relation will be implicitly modeled by skip-connect relation between nodes.

The model recurrently updates the hidden states according to:

$$m_t = h(x_t, m_0, ..., m_{t-1}, g_0^t, ..., g_{t-1}^t) \tag{2}$$

and the probability distribution for next word is approximated by:

$$p(x_{t+1}|x_0, ..., x_t) \approx p(x_{t+1}; f(m_0, ..., m_t, g_0^{t+1}, ..., g_t^{t+1})) \tag{3}$$

where $g_i^t$ are gates that control skip-connections. Both $f$ and $h$ have a structured attention mechanism that takes $g_i^t$ as input and forces the model to focus on the most related information. Since $l_t$ is an unobserved latent variable, We explain an approximation for $g_i^t$ in the next section. The structured attention mechanism is explained in section 5.1.

## 4 MODELING SYNTACTIC STRUCTURE

### 4.1 MODELING LOCAL STRUCTURE

In this section we give a probabilistic view on how to model the local structure of language. A detailed elaboration for this section is given in Appendix B. At time step $t$, $p(l_t|x_0, ..., x_t)$ represents the probability of choosing one out of $t$ possible local structures. We propose to model the distribution by the Stick-Breaking Process:

$$p(l_t = i|x_0, ..., x_t) = (1 - \alpha_i^t) \prod_{j=i+1}^{t-1} \alpha_j^t \tag{4}$$

The formula can be understood by noting that after the time step $i+1, ..., t-1$ have their probabilities assigned, $\prod_{j=i+1}^{t-1} \alpha_j^t$ is remaining probability, $1 - \alpha_i^t$ is the portion of remaining probability that we assign to time step $i$. Variable $\alpha_j^t$ is parametrized in the next section.

As shown in Appendix B, the expectation of gate value $g_i^t$ is the Cumulative Distribution Function (CDF) of $p(l_t = i|x_0, ..., x_t)$. Thus, we can replace the discrete gate value by its expectation:

$$g_i^t = \mathbf{P}(l_t \leq i) = \prod_{j=i+1}^{t-1} \alpha_j^t \tag{5}$$

With these relaxations, Eq.2 and 3 can be approximated by using a soft gating vector to update the hidden state and predict the next token.

### 4.2 PARSING NETWORK

**Inferring tree structure with Syntactic Distance**    In Eq.4, $1 - \alpha_j^t$ is the portion of the remaining probability that we assign to position $j$. Because the stick-breaking process should assign high probability to $l_t$, which is the closest constituent-beginning word. The model should assign large $1 - \alpha_j^t$ to words beginning new constituents. While $x_t$ itself is a constituent-beginning word, the model should assign large $1 - \alpha_j^t$ to words beginning larger constituents. In other words, the model will consider longer dependency relations for the first word in constituent. Given the sentence in Figure 1, at time step $t = 6$, both $1 - \alpha_2^6$ and $1 - \alpha_0^6$ should be close to 1, and all other $1 - \alpha_j^6$ should be close to 0.

In order to parametrize $\alpha_j^t$, our basic hypothesis is that words in the same constituent should have a closer syntactic relation within themselves, and that this syntactical proximity can be represented by a scalar value. From the tree structure point of view, the shortest path between leafs in same subtree is shorter than the one between leafs in different subtree.

To model syntactical proximity, we introduce a new feature *Syntactic Distance*. For a sentence with length $K$, we define a set of $K$ real valued scalar variables $d_0, ..., d_{K-1}$, with $d_i$ representing a measure of the syntactic relation between the pair of adjacent words $(x_{i-1}, x_i)$. $x_{-1}$ could be the last word in previous sentence or a padding token. For time step $t$, we want to find the closest words $x_j$, that have larger syntactic distance than $d_t$. Thus $\alpha_j^t$ can be defined as:

$$\alpha_j^t = \frac{\text{hardtanh}\left((d_t - d_j) \cdot \tau\right) + 1}{2} \tag{6}$$

where $\text{hardtanh}(x) = \max(-1, \min(1, x))$. $\tau$ is the temperature parameter that controls the sensitivity of $\alpha_j^t$ to the differences between distances.

The Syntactic Distance has some nice properties that both allow us to infer a tree structure from it and be robust to intermediate non-valid tree structures that the model may encounter during learning. In Appendix C and D we list these properties and further explain the meanings of their values.

**Parameterizing Syntactic Distance**    Roark & Hollingshead (2008) shows that it's possible to identify the beginning and ending words of a constituent using local information. In our model, the syntactic distance between a given token (which is usually represented as a vector word embedding $e_i$) and its previous token $e_{i-1}$, is provided by a convolutional kernel over a set of consecutive previous tokens $e_{i-L}, e_{i-L+1}, ..., e_i$. This convolution is depicted as the gray triangles shown in Figure 3. Each triangle here represent 2 layers of convolution. Formally, the syntactic distance $d_i$ between token $e_{i-1}$ and $e_i$ is computed by

$$h_i = \text{ReLU}(W_c \begin{bmatrix} e_{i-L} \\ e_{i-L+1} \\ ... \\ e_i \end{bmatrix} + b_c) \tag{7}$$

$$d_i = \text{ReLU}\left(W_d h_i + b_d\right) \tag{8}$$

where $W_c$, $b_c$ are the kernel parameters. $W_d$ and $b_d$ can be seen as another convolutional kernel with window size 1, convolved over $h_i$'s. Here the kernel window size $L$ determines how far back into the history node $e_i$ can reach while computing its syntactic distance $d_i$. Thus we call it the *look-back range*.

Convolving $\boldsymbol{h}$ and $\boldsymbol{d}$ on the whole sequence with length $K$ yields a set of distances. For the tokens in the beginning of the sequence, we simply pad $L - 1$ zero vectors to the front of the sequence in order to get $K - 1$ outputs.

## 5 MODELING LANGUAGE

### 5.1 READING NETWORK

The Reading Network generate new states $m_t$ considering on input $x_t$, previous memory states $m_0, ..., m_{t-1}$, and gates $g_0^t, ..., g_{t-1}^t$, as shown in Eq.2.

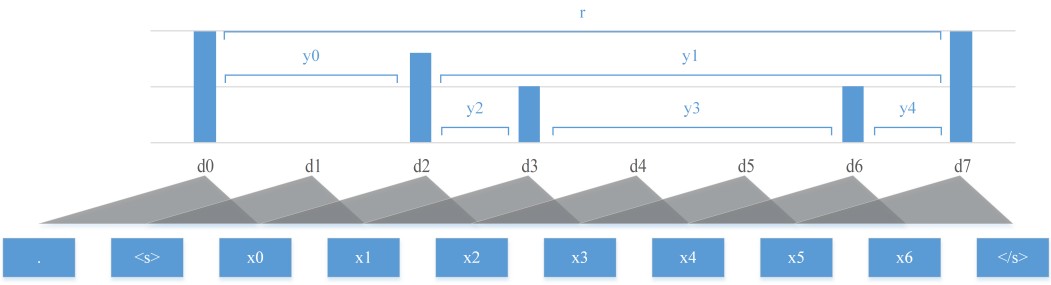

Figure 3: Convolutional network for computing syntactic distance. Gray triangles represent 2 layers of convolution, $d_0$ to $d_7$ are the syntactic distance output by each of the kernel position. The blue bars indicate the amplitude of $d_i$'s, and $y_i$'s are the inferred constituents.

Similar to Long Short-Term Memory-Network (LSTMN) (Cheng et al., 2016), the Reading Network maintains the memory states by maintaining two sets of vectors: a hidden tape $H_{t-1} = (h_{t-N_m}, ..., h_{t-1})$, and a memory tape $C_{t-1} = (c_{t-L}, ..., c_{t-1})$, where $N_m$ is the upper bound for the memory span. Hidden states $m_i$ is now represented by a tuple of two vectors $(h_i, c_i)$. The Reading Network captures the dependency relation by a modified attention mechanism: *structured attention*. At each step of recurrence, the model summarizes the previous recurrent states via the structured attention mechanism, then performs a normal LSTM update, with hidden and cell states output by the attention mechanism.

**Structured Attention**  At each time step $t$, the *read* operation attentively links the current token to previous memories with a structured attention layer:

$$k_t = W_h h_{t-1} + W_x x_t \tag{9}$$

$$\tilde{s}_i^t = \text{softmax}\left(\frac{h_i k_t^{\text{T}}}{\sqrt{\delta_k}}\right) \tag{10}$$

where, $\delta_k$ is the dimension of the hidden state. Modulated by the gates in Eq.5, the structured intra-attention weight is defined as:

$$s_i^t = \frac{g_i^t \tilde{s}_i^t}{\sum_i g_i^t} \tag{11}$$

This yields a probability distribution over the hidden state vectors of previous tokens. We can then compute an adaptive summary vector for the previous hidden tape and memory denoting by $\tilde{h}_t$ and $\tilde{c}_t$:

$$\begin{bmatrix} \tilde{h}_t \\ \tilde{c}_t \end{bmatrix} = \sum_{i=1}^{t-1} s_i^t \cdot m_i = \sum_{i=1}^{t-1} s_i^t \cdot \begin{bmatrix} h_i \\ c_i \end{bmatrix} \tag{12}$$

Structured attention provides a way to model the dependency relations shown in Figure 1.

**Recurrent Update**  The Reading Network takes $x_t$, $\tilde{c}_t$ and $\tilde{h}_t$ as input, computes the values of $c_t$ and $h_t$ by the LSTM recurrent update (Hochreiter & Schmidhuber, 1997). Then the *write* operation concatenates $h_t$ and $c_t$ to the end of hidden and memory tape.

## 5.2  PREDICT NETWORK

Predict Network models the probability distribution of next word $x_{t+1}$, considering on hidden states $m_0, ..., m_t$, and gates $g_0^{t+1}, ..., g_t^{t+1}$. Note that, at time step $t$, the model cannot observe $x_{t+1}$, a temporary estimation of $d_{t+1}$ is computed considering on $x_{t-L}, ..., x_t$:

$$d_{t+1}' = \text{ReLU}(W_d' h_t + b_d') \tag{13}$$

From there we compute its corresponding $\{\alpha^{t+1}\}$ and $\{g_i^{t+1}\}$ for Eq.3. We parametrize $f(\cdot)$ function as:

$$f(m_0, ..., m_t, g_0^{t+1}, ..., g_t^{t+1}) = \hat{f}([h_{l:t-1}, h_t]) \tag{14}$$

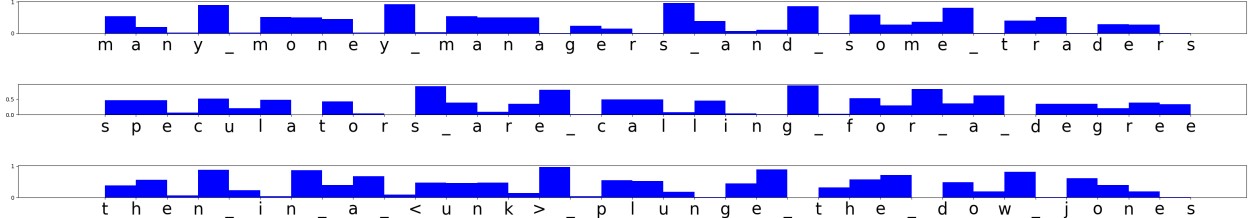

Figure 4: Syntactic distance estimated by Parsing Network. The model is trained on PTB dataset at the character level. Each blue bar is positioned between two characters, and represents the syntactic distance between them. From these distances we can infer a tree structure according to Section 4.2.

where $h_{l:t-1}$ is an adaptive summary of $h_{l_{t+1} \leq i \leq t-1}$, output by structured attention controlled by $g_0^{t+1}, ..., g_{t-1}^{t+1}$. $\hat{f}(\cdot)$ could be a simple feed-forward MLP, or more complex architecture, like ResNet, to add more depth to the model.

## 6 EXPERIMENTS

We evaluate the proposed model on three tasks, character-level language modeling, word-level language modeling, and unsupervised constituency parsing.

### 6.1 CHARACTER-LEVEL LANGUAGE MODEL

From a character-level view, natural language is a discrete sequence of data, where discrete symbols form a distinct and shallow tree structure: the sentence is the root, words are children of the root, and characters are leafs. However, compared to word-level language modeling, character-level language modeling requires the model to handle longer-term dependencies. We evaluate a character-level variant of our proposed language model over a preprocessed version of the Penn Treebank (PTB) and Text8 datasets.

When training, we use truncated back-propagation, and feed the final memory position from the previous batch as the initial memory of next one. At the beginning of training and test time, the model initial hidden states are filled with zero. Optimization is performed with Adam using learning rate $lr = 0.003$, weight decay $w_{decay} = 10^{-6}$, $\beta_1 = 0.9$, $\beta_2 = 0.999$ and $\sigma = 10^{-8}$. We carry out gradient clipping with maximum norm 1.0. The learning rate is multiplied by 0.1 whenever validation performance does not improve during 2 checkpoints. These checkpoints are performed at the end of each epoch. We also apply layer normalization (Ba et al., 2016) to the Reading Network and batch normalization to the Predict Network and parsing network. For all of the character-level language modeling experiments, we apply the same procedure, varying only the number of hidden units, mini-batch size and dropout rate.

**Penn Treebank** we process the Penn Treebank dataset (Marcus et al., 1993) by following the procedure introduced in (Mikolov et al., 2012). For character-level PTB, Reading Network has two recurrent layers, Predict Network has one residual block. Hidden state size is 1024 units. The input and output embedding size are 128, and not shared. Look-back range $L = 10$, temperature parameter $\tau = 10$, upper band of memory span $N_m = 20$. We use a batch size of 64, truncated back-propagation with 100 timesteps. The values used of dropout on input/output embeddings, between recurrent layers, and on recurrent states were (0, 0.25, 0.1) respectively.

In Figure 4, we visualize the syntactic distance estimated by the Parsing Network, while reading three different sequences from the PTB test set. We observe that the syntactic distance tends to be higher between the last character of a word and a space, which is a reasonable breakpoint to separate between words. In other words, if the model sees a space, it will attend on all previous step. If the model sees a letter, it will attend no further then the last space step. The model autonomously discovered to avoid inter-word attention connection, and use the hidden states of space (separator) tokens to summarize previous information. This is strong proof that the model can understand the latent structure of data. As a result our model achieve state-of-the-art performance and significantly

| Model | BPC |
|---|---|
| Norm-stabilized RNN (Krueger & Memisevic, 2015) | 1.48 |
| CW-RNN (Koutnik et al., 2014) | 1.46 |
| HF-MRNN (Mikolov et al., 2012) | 1.41 |
| MI-RNN (Wu et al., 2016) | 1.39 |
| ME n-gram (Mikolov et al., 2012) | 1.37 |
| BatchNorm LSTM (Cooijmans et al., 2016) | 1.32 |
| Zoneout RNN (Krueger et al., 2016) | 1.27 |
| HyperNetworks (Ha et al., 2016) | 1.27 |
| LayerNorm HM-LSTM (Chung et al., 2016) | 1.24 |
| LayerNorm HyperNetworks (Ha et al., 2016) | 1.23 |
| PRPN | **1.202** |

Table 1: BPC on the Penn Treebank test set

outperform baseline models. It is worth noting that HM-LSTM (Chung et al., 2016) also unsupervisedly induce similar structure from data. But discrete operations in HM-LSTM make their training procedure more complicated then ours.

## 6.2 WORD-LEVEL LANGUAGE MODEL

Comparing to character-level language modeling, word-level language modeling needs to deal with complex syntactic structure and various linguistic phenomena. But it has less long-term dependencies. We evaluate the word-level variant of our language model on a preprocessed version of the Penn Treebank (PTB) (Marcus et al., 1993) and Text8 (Mahoney, 2011) dataset.

We apply the same procedure and hyper-parameters as in character-level language model. Except optimization is performed with Adam with $\beta_1 = 0$. This turns off the exponential moving average for estimates of the means of the gradients (Melis et al., 2017). We also adapt the number of hidden units, mini-batch size and the dropout rate according to the different tasks.

**Penn Treebank**   we process the Penn Treebank dataset (Mikolov et al., 2012) by following the procedure introduced in (Mikolov et al., 2010). For word-level PTB, the Reading Network has two recurrent layers and the Predict Network do not have residual block. The hidden state size is 1200 units and the input and output embedding sizes are 800, and shared (Inan et al., 2016; Press & Wolf, 2017). Look-back range $L = 5$, temperature parameter $\tau = 10$ and the upper band of memory span $N_m = 15$. We use a batch size of 64, truncated back-propagation with 35 time-steps. The values used of dropout on input/output embeddings, between recurrent layers, and on recurrent states were (0.7, 0.5, 0.5) respectively.

| Model | PPL |
|---|---|
| RNN-LDA + KN-5 + cache (Mikolov & Zweig, 2012) | 92.0 |
| LSTM (Zaremba et al., 2014) | 78.4 |
| Variational LSTM (Kim et al., 2016) | 78.9 |
| CharCNN (Kim et al., 2016) | 78.9 |
| Pointer Sentinel-LSTM (Merity et al., 2016) | 70.9 |
| LSTM + continuous cache pointer (Grave et al., 2016) | 72.1 |
| Variational LSTM (tied) + augmented loss (Inan et al., 2016) | 68.5 |
| Variational RHN (tied) (Zilly et al., 2016) | 65.4 |
| NAS Cell (tied) (Zoph & Le, 2016) | 62.4 |
| 4-layer skip connection LSTM (tied) (Melis et al., 2017) | **58.3** |
| PRPN | 61.98 |

Table 2: PPL on the Penn Treebank test set

| Model | PPL |
|---|---|
| PRPN | 61.98 |
| - Parsing Net | 64.42 |
| - Reading Net Attention | 64.63 |
| - Predict Net Attention | 63.65 |
| Our 2-layer LSTM | 65.81 |

Table 3: Ablation test on the Penn Treebank. "- Parsing Net" means that we remove Parsing Network and replace Structured Attention with normal attention mechanism; "- Reading Net Attention" means that we remove Structured Attention from Reading Network, that is equivalent to replace Reading Network with a normal 2-layer LSTM; "- Predict Net Attention" means that we remove Structured Attention from Predict Network, that is equivalent to have a standard projection layer; "Our 2-layer LSTM" is equivalent to remove Parsing Network and remove Structured Attention from both Reading and Predict Network.

**Text8** dataset contains 17M training tokens and has a vocabulary size of 44k words. The dataset is partitioned into a training set (first 99M characters) and a development set (last 1M characters) that is used to report performance. As this dataset contains various articles from Wikipedia, the longer term information (such as current topic) plays a bigger role than in the PTB experiments (Mikolov et al., 2014). We apply the same procedure and hyper-parameters as in character-level PTB, except we use a batch size of 128. The values used of dropout on input/output embeddings, between Recurrent Layers and on recurrent states were (0.4, 0.2, 0.2) respectively.

| Model | PPL |
|---|---|
| LSTM-500 (Mikolov et al., 2014) | 156 |
| SCRNN (Mikolov et al., 2014) | 161 |
| MemNN (Sukhbaatar et al., 2015) | 147 |
| LSTM-1024 (Grave et al., 2016) | 121 |
| LSTM + continuous cache pointer (Grave et al., 2016) | 99.9 |
| PRPN | **81.64** |

Table 4: PPL on the Text8 valid set

In Table 2, our results are comparable to the state-of-the-art methods. Since we do not have the same computational resource used in (Melis et al., 2017) to tune hyper-parameters at large scale, we expect that our model could achieve better performance after an aggressive hyperparameter tuning process. As shown in Table 4, our method outperform baseline methods. It is worth noticing that the *continuous cache pointer* can also be applied to output of our Predict Network without modification. Visualizations of tree structure generated from learned PTB language model are included in Appendix A. In Table 3, we show the value of test perplexity for different variants of PRPN, each variant remove part of the model. By removing Parsing Network, we observe a significant drop of performance. This stands as empirical evidence regarding the benefit of having structure information to control attention.

## 6.3 UNSUPERVISED CONSTITUENCY PARSING

The unsupervised constituency parsing task compares hte tree structure inferred by the model with those annotated by human experts. The experiment is performed on WSJ10 dataset. WSJ10 is the 7422 sentences in the Penn Treebank Wall Street Journal section which contained 10 words or less after the removal of punctuation and null elements. Evaluation was done by seeing whether proposed constituent spans are also in the Treebank parse, measuring unlabeled F1 ($UF_1$) of unlabeled constituent precision and recall. Constituents which could not be gotten wrong (those of span one and those spanning entire sentences) were discarded. Given the mechanism discussed in Section 4.2, our model generates a binary tree. Although standard constituency parsing tree is not limited to binary tree. Previous unsupervised constituency parsing model also generate binary trees (Klein

& Manning, 2002; Bod, 2006). Our model is compared with the several baseline methods, that are explained in Appendix E.

Different from the previous experiment setting, the model treat each sentence independently during train and test time. When training, we feed one batch of sentences at each iteration. In a batch, shorter sentences are padded with 0. At the beginning of the iteration, the model's initial hidden states are filled with zero. When testing, we feed on sentence one by one to the model, then use the gate value output by the model to recursively combine tokens into constituents, as described in Appendix A.

| Model | $UF_1$ |
|---|---|
| LBRANCH | 28.7 |
| RANDOM | 34.7 |
| DEP-PCFG (Carroll & Charniak, 1992) | 48.2 |
| RBRANCH | 61.7 |
| CCM (Klein & Manning, 2002) | 71.9 |
| DMV+CCM (Klein & Manning, 2005) | 77.6 |
| UML-DOP (Bod, 2006) | **82.9** |
| PRPN | 70.02 |
| UPPER BOUND | 88.1 |

Table 5: Parsing Performance on the WSJ10 dataset

Table 5 summarizes the results. Our model significantly outperform the RANDOM baseline indicate a high consistency with human annotation. Our model also shows a comparable performance with CCM model. In fact our parsing network and CCM both focus on the relation between successive tokens. As described in Section 4.2, our model computes syntactic distance between all successive pair of tokens, then our parsing algorithm recursively assemble tokens into constituents according to the learned distance. CCM also recursively model the probability whether a contiguous subsequences of a sentence is a constituent. Thus, one can understand how our model is outperformed by DMV+CCM and UML-DOP models. The DMV+CCM model has extra information from a dependency parser. The UML-DOP approach captures both contiguous and non-contiguous lexical dependencies (Bod, 2006).

## 7 CONCLUSION

In this paper, we propose a novel neural language model that can simultaneously induce the syntactic structure from unannotated sentences and leverage the inferred structure to learn a better language model. We introduce a new neural parsing network: Parsing-Reading-Predict Network, that can make differentiable parsing decisions. We use a new structured attention mechanism to control skip connections in a recurrent neural network. Hence induced syntactic structure information can be used to improve the model's performance. Via this mechanism, the gradient can be directly back-propagated from the language model loss function into the neural Parsing Network. The proposed model achieve (or is close to) the state-of-the-art on both word/character-level language modeling tasks. Experiment also shows that the inferred syntactic structure highly correlated to human expert annotation.

## ACKNOWLEDGEMENT

The authors would like to thank Timothy J. O'Donnell and Chris Dyer for the helpful discussions.

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

# APPENDIX

## A    INFERRED TREE STRUCTURE

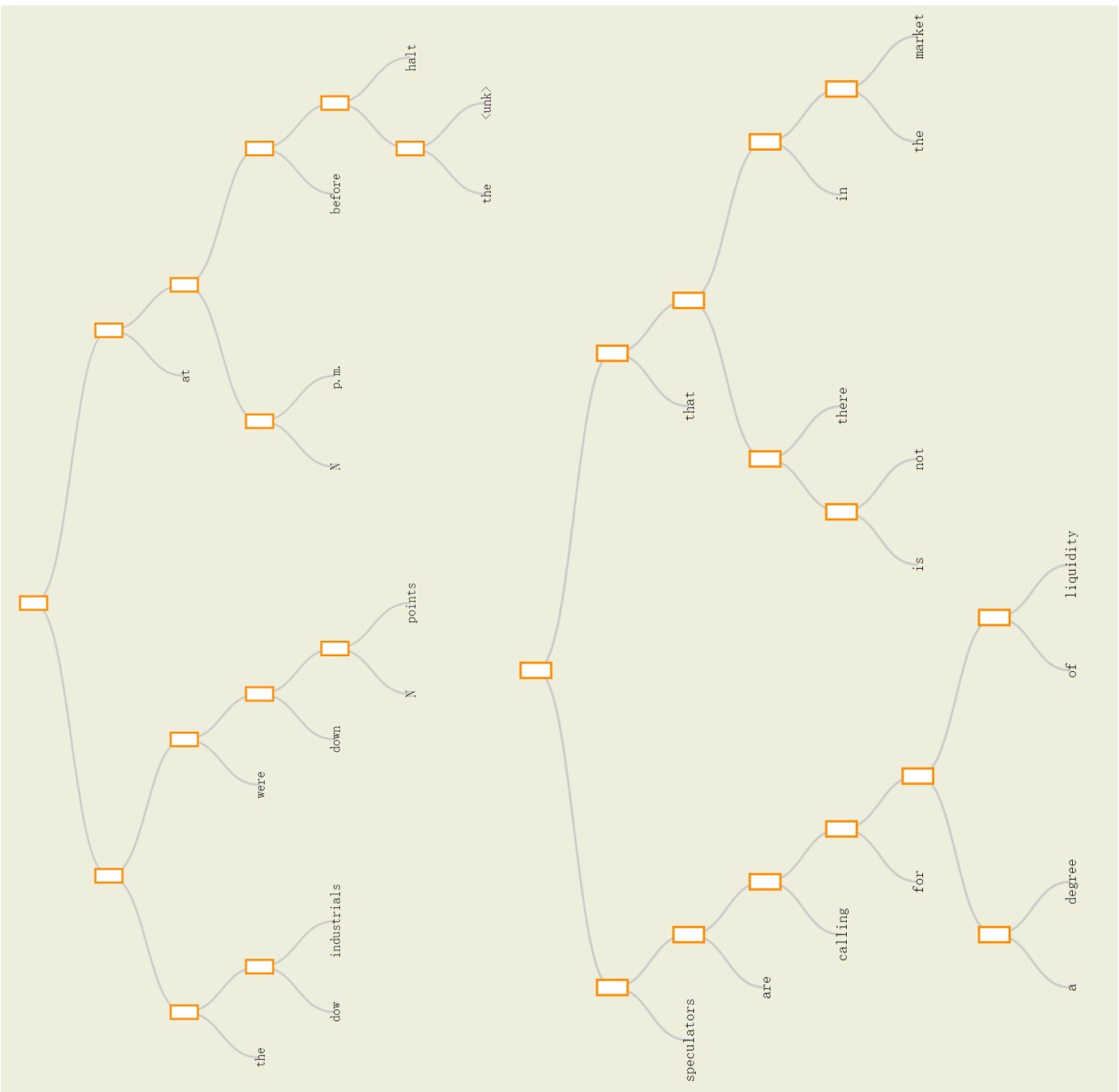

Figure 5: Syntactic structures of two different sentences inferred from $\{d_i\}$ given by Parsing Network.

The tree structure is inferred from the syntactic distances yielded by the Parsing Network. We first sort the $d_i$'s in decreasing order. For the first $d_i$ in the sorted sequence, we separate sentence into constituents $((x_{<i}), (x_i, (x_{>i})))$. Then we separately repeat this operation for constituents $(x_{<i})$ and $(x_{>i})$. Until the constituent contains only one word.

# B  MODELING LOCAL STRUCTURE

In this section we give a probabilistic view on how to model the local structure of language. Given the nature of language, sparse connectivity can be enforced as a prior on how to improve generalization and interpretability of the model.

At time step $t$, $p(l_t|x_0, ..., x_t)$ represents the probability of choosing one out of $t$ possible local structures that defines the conditional dependencies. If $l_t = t'$, it means $x_t$ depends on all the previous hidden state from $m_{t'}$ to $m_t$ ($t' \leq t$).

A particularly flexible option for modeling $p(l_t|x_0, ..., x_t)$ is the Dirichlet Process, since being nonparametric allows us to attend on as many words as there are in a sentence; i.e. number of possible structures (mixture components) grows with the length of the sentence. As a result, we can write the probability of $l_{t+1} = t'$ as a consequence of the stick breaking process [1]:

$$p(l_t = t'|x_0, ..., x_t) = (1 - \alpha_{t'}^t) \prod_{j=t'+1}^{t-1} \alpha_j^t \tag{15}$$

for $1 \leq t' < t - 1$, and

$$p(l_t = t-1|x_0, ..., x_t) = (1 - \alpha_{t-1}^t); \qquad p(l_t = 0|x_0, ..., x_t) = \prod_{j=1}^{t-1} \alpha_j^t \tag{16}$$

where $\alpha_j = 1 - \beta_j$ and $\beta_j$ is a sample from a Beta distribution. Once we sample $l_t$ from the process, the connectivity is realized by a element-wise multiplication of an attention weight vector with a masking vector $g_t$ defined in Eq. 1. In this way, $x_t$ becomes functionally independent of all $x_s$ for all $s < l_t$. The expectation of this operation is the CDF of the probability of $l$, since

$$\mathbf{E}_{l_t}[g_t^{\{t'\}}] = \prod_{j=1}^{t} \alpha_j^t + (1 - \alpha_1^t) \prod_{j=2}^{t} \alpha_j^t + ... + (1 - \alpha_{t'}^t) \prod_{j=t'+1}^{t} \alpha_j^t$$
$$= \sum_{k=0}^{t'} p(l_t = k|x_0, ..., x_t) = \mathbf{P}(l_t \leq t') \tag{17}$$

By telescopic cancellation, the CDF can be expressed in a succinct way:

$$\mathbf{P}(l_t \leq t') = \prod_{j=t'+1}^{t-1} \alpha_j^t \tag{18}$$

for $t' < t$, and $\mathbf{P}(l_t \leq t) = 1$. However, being Bayesian nonparametric and assuming a latent variable model require approximate inference. Hence, we have the following relaxations

1. First, we relax the assumption and parameterize $\alpha_j^t$ as a deterministic function depending on all the previous words, which we will describe in the next section.

2. We replace the discrete decision on the graph structure with a soft attention mechanism, by multiplying attention weight with the multiplicative gate:

$$g_i^t = \prod_{j=i+1}^{t} \alpha_j^t \tag{19}$$

With these relaxations, Eq. (3) can be approximated by using a soft gating vector to update the hidden state $h$ and the predictive function $f$. This approximation is reasonable since the gate is the expected value of the discrete masking operation described above.

---

[1]Note that the index is in decreasing order.

## C  No Partial Overlapping in Dependency Ranges

In this appendix, we show that having no partial overlapping in dependency ranges is an essential property for recovering a valid tree structure, and PRPN can provide a binary version of $g_i^t$, that have this property.

The masking vector $g_i^t$ introduced in Section 4.1 determines the range of dependency, i.e., for the word $x_t$ we have $g_i^t = 1$ for all $l_t \leq i < t$. All the words fall into the range $l_t \leq i < t$ is considered as $x_t$'s sibling or offspring of its sibling. If the dependency ranges of two words are disjoint with each other, that means the two words belong to two different subtrees. If one range contains another, that means the one with smaller range is a sibling, or is an offspring of a sibling of the other word. However, if they partially overlaps, they can't form a valid tree.

While Eq.5 and Eq.6 provide a soft version of dependency range, we can recover a binary version by setting $\tau$ in Eq.6 to $+\infty$. The binary version of $\alpha_j^t$ corresponding to Eq. 6 becomes:

$$\alpha_j^t = \frac{\text{sign}\,(d_t - d_{j+1}) + 1}{2} \tag{20}$$

which is basically the sign of comparing $d_t$ and $d_{j+1}$, scaled to the range of 0 and 1. Then for each of its previous token the *gate* value $g_i^t$ can be computed through Eq.5.

Now for a certain $x_t$, we have

$$g_i^t = \begin{cases} 1, & t' \leq i < t \\ 0, & 0 \leq i < t' \end{cases} \tag{21}$$

where

$$t' = \max i, \quad s.t. \quad d_i > d_t \tag{22}$$

Now all the words that fall into the range $t' \leq i < t$ are considered as either sibling of $x_t$, or offspring of a sibling of $x_t$ (Figure 3). The essential point here is that, under this parameterization, the dependency range of any two tokens won't partially overlap. Here we provide a terse proof:

*Proof.* Let's assume that the dependency range of $x_v$ and $x_n$ partially overlaps. We should have $g_i^u = 1$ for $u \leq i < v$ and $g_i^n = 1$ for $m \leq i < n$. Without losing generality, we assume $u < m < v < n$ so that the two dependency ranges overlap in the range $[m, v]$.

1. For $x_v$, we have $\alpha_i^v = 1$ for all $u \leq i < v$. According to Eq. 6 and 5, we have $d_i < d_v$ for all $u \leq i < v$. Since $u < m$, we have $d_m < d_v$.

2. Similarly, for $x_n$, we have $d_i < d_n$ for all $m \leq i < n$. Since $m < v$, we have $d_v < d_n$. On the other hand, since the range stops at $m$, we should also have $d_m > d_n$. Thus $d_m > d_v$.

Items 1 and 2 are contradictory, so the dependency ranges of $x_v$ and $x_n$ won't partially overlap. $\square$

## D  Properties and Intuitions of $g_i^t$ and $d_i$

First, for any fixed $t$, $g_i^t$ is monotonic in $i$. This ensures that $g_i^t$ still provides soft truncation to define a dependency range.

The second property comes from $\tau$. The hyperparameter $\tau$ has an interesting effect on the tree structure: if it is set to 0, then for all $t$, the gates $g_i^t$ will be open to all of $e_t$'s predecessors, which will result in a flat tree where all tokens are direct children of the root node; as $\tau$ becomes larger, the number of levels of hierarchy in the tree increases. As it approaches $+\inf$, the $\text{hardtanh}(\cdot)$ becomes $\text{sign}(\cdot)$ and the dependency ranges form a valid tree. Note that, due to the linear part of the gating mechanism, which benefits training, when $\tau$ has a value in between the two extremes the truncation range for each token may overlap. That may sometimes result in vagueness in some part of the inferred tree. To eliminate this vagueness and ensure a valid tree, at test time we use $\tau = +\inf$.

Under this framework, the values of syntactic distance have more intuitive meanings. If two adjacent words are siblings of each other, the syntactic distance should approximate zero; otherwise, if they

belong to different subtrees, they should have a larger syntactic distance. In the extreme case, the syntactic distance approaches 1 if the two words have no subtree in common. In Figure 3 we show the syntactic distances for each adjacent token pair which results in the tree shown in Figure 1.

## E    BASELINE METHODS FOR UNSUPERVISED CONSTITUENCY PARSING

Our model is compared with the same baseline methods as in (Klein & Manning, 2005). *RANDOM* chooses a binary tree uniformly at random from the set of binary trees. This is the unsupervised baseline. *LBRANCH* and *RBRANCH* choose the completely left- and right-branching structures, respectively. *RBRANCH* is a frequently used baseline for supervised parsing, but it should be stressed that it encodes a significant fact about English structure, and an induction system need not beat it to claim a degree of success. *UPPER BOUND* is the upper bound on how well a binary system can do against the Treebank sentences. Because the Treebank sentences are generally more flat than binary, limiting the maximum precision which can be attained, since additional brackets added to provide a binary tree will be counted as wrong.

We also compared our model with other unsupervised constituency parsing methods. *DEP-PCFG* is dependency-structured PCFG (Carroll & Charniak, 1992). *CCM* is constituent-context model (Klein & Manning, 2002). *DMV* is an unsupervised dependency parsing model. *DMV+CCM* is a combined model that jointly learn both constituency and dependency parser (Klein & Manning, 2004).

