# OpenReview forum: "Neural Language Modeling by Jointly Learning Syntax and Lexicon"
_ICLR.cc/2018/Conference — Accept (Poster)_

### Official Review · AnonReviewer3 · 2017-11-27

**Rating:** 8
**Confidence:** 4

**Review:**

Summary: the paper proposes a novel method to leverage tree structures in an unsupervised learning manner. The key idea is to make use of “syntactic distance” to identify phrases, thus building up a tree for input sentence. The proposed model achieves SOTA on a char-level language modeling task and is demonstrated to yield reasonable tree structures.

Comment: I like the paper a lot. The idea is very creative and interesting. The paper is well written.

Besides the official comment that the authors already replied, I have some more:
- I was still wondering how to compute the left hand side of eq 3 by marginalizing over all possible unfinished structures so far. (Of course, what the authors do is showed to be a fast and good approximation.)
- Using CNN to compute d has a disadvantage that the range of look-back must be predefined. Looking at fig 3, in order to make sure that d6 is smaller than d2, the look-back should have a wide coverage so that the computation for d6 has some knowledge about d2 (in some cases the local information can help to avoid it, but not always). I therefore think that using an RNN is more suitable than using a CNN.
- Is it possible to extend this framework to dependency structure?
- It would be great if the authors show whether the model can leverage given tree structures (like SPINN) (for instance we can do a multitask learning where a task is parsing given a treebank to train)

---

> ### Author Response · Authors · 2017-12-23
> **Responses to Reviewer3**
>
> Thanks a lot for your kind review and suggestions. We’d like to address your issues as follows:
>
> Regarding "marginalizing over all possible unfinished structures so far"
> Marginalizing over all possible unfinished structure is very difficult due to the fact that our model stacks multiple recurrent layers. One better approximation is that we compute left-hand side of both eq2 and eq3 by marginalizing over all possible local structures at each time step. In other words, we can sampling all possible l_t, then compute the weighted sum of the right-hand side of eq2 and eq3 with respect to different l_t and using p(l_{t}=t'|x_0, ...,x_t) as weights.
>
> Regarding "using an RNN is more suitable than using a CNN"
> We totally agree with that. Using an LSTM can provide an unbounded context information for the gates, and that is definitely a good direction to try. We will probably try that in the future iterations of our model.
>
> Regarding "extend this framework to dependency structure"
> Parsing network can only give boundary information for constituent parsing. However, it's possible to extract dependency information from attention weights, which remains an open question to study.
>
> Regarding "leverage given tree structures"
> We also have thought about this. One possible way is to infer a set of true distances using the given tree structure and train the parsing network to generate a set of distances which align with the true distances. We haven’t done that in this work since we want to focus on unsupervised learning. This will be explored in our next work.
>
> Thanks again for the comments and review!

---

### Official Review · AnonReviewer1 · 2017-11-27

**Rating:** 7
**Confidence:** 3

**Review:**

** UPDATE ** upgraded my score to 7 based on the new version of the paper.

The main contribution of this paper is to introduce a new recurrent neural network for language modeling, which incorporates a tree structure More precisely, the model learns constituency trees (without any supervision), to capture syntactic information. This information is then used to define skip connections in the language model, to capture longer dependencies between words. The update of the hidden state does not depend only on the previous hidden state, but also on the hidden states corresponding to the following words: all the previous words belonging to the smallest subtree containing the current word, such that the current word is not the left-most one. The authors propose to parametrize trees using "syntactic distances" between adjacent words (a scalar value for each pair of adjacent words w_t, w_{t+1}). Given these distances, it is possible to obtain the constituents and the corresponding gating activations for the skip connections. These different operations can be relaxed to differentiable operations, so that stochastic gradient descent can be used to learn the parameters. The model is evaluated on three language modeling benchmarks: character level PTB, word level PTB and word level text8. The induced constituency trees are also evaluated, for sentence of length 10 or less (which is the standard setting for unsupervised parsing).

Overall, I really like the main idea of the paper. The use of "syntactic distances" to parametrize the trees is clever, as they can easily be computed using only partial information up to time t. From these distances, it is also relatively straightforward to obtain which constituents (or subtrees) a word belongs to (and thus, the corresponding gating activations). Moreover, the operations can easily be relaxed to obtain a differentiable model, which can easily be trained using stochastic gradient descent.

The results reported on the language modeling experiments are strong. One minor comment here is that it would be nice to have an ablation analysis, as it is possible to obtain similarly strong results with simpler models (such as plain LSTM).

My main concern regarding the paper is that it is a bit hard to understand. In particular in section 4, the authors alternates between discrete and relaxed values: end of section 4.1, it is implied that alpha are in [0, 1], but in equation 6, alpha are in {0, 1}, then relaxed in equation 9 to [0, 1] again. I am also wondering whether it would make more sense to start by introducing the syntactic distances, then the alphas and finally the gates? I also found the section 5 to be quite confusing. While I get the	general idea, I am not sure what is the relation between hidden states h and m (section 5.1). Is there a mixup between h defined in equation 10 and h from section 5.1? I am aware that it is not straightforward to describe the proposed method, but believe it would be a much stronger paper if written more clearly.

To conclude, I really like the method proposed in this paper, and believe that the experimental results are quite strong.
My main concern	regarding the paper is its clarity: I will gladly increase my score if the authors can improve the writing.

---

> ### Author Response · Authors · 2017-12-23
> **Responses to Reviewer1**
>
> Thanks for the comments and suggestions. We have modified our manuscript accordingly in the updated version of the paper.
>
> For the ablation studies, we’ve added a set of results in Section 6.2, Table 3.
>
> We are sorry for the lack of clarity in the paper, and we have largely rewritten Section 4 in the hope of clarifying our explanation.
>
> To answer the question in the review, \alpha is expected to be in [0, 1] throughout the paper. In Eq. 6 in the updated paper, the hardtanh() function is a piecewise linear function defined by hardtanh(x) = max(-1, min(1, x)), which has a linear slope near zero, so its output is also in [0, 1]. In section 5.1, the m is the state that we regard as memory. In the case of using an LSTM, which is what we are doing in the experiments, we are modifying both h and c according to the attention weights, so m=(h, c). In Eq. 10, h stands for the hidden states only. We modified Section 5.1 to make these differences between h and m clearer.
>
> Thanks again for your precious comments!

---

### Official Review · AnonReviewer2 · 2017-11-30
**solid experiments and interesting model**

**Rating:** 7
**Confidence:** 4

**Review:**

The paper proposes Parsing-Reading-Predict Networks (PRPN), a new model jointly learns syntax and lexicon. The main idea of this model is to add skip-connections to integrate syntax relationships into the context of predicting the next word (i.e. language modeling task).

To model this, the authors introduce hidden variable l_t, which break down to the decisions of a soft version of gate variable values in the previous possible positions. These variables are then parameterized using syntactic distance to ensure that the final structure inferred by the model has no overlapping ranges so that it will be a valid syntax tree.

I think the paper is in general clearly written. The model is interesting and the experiment section is quite solid. The model reaches state-of-the-art level performance in language modeling and the performance on unsupervised parsing task (which is a by-product of the model) is also quite promising.

My main question is that the motivation/intuition of introducing the syntactic distance variable. I understand that they basically make sure the tree is valid, but the paper did not explain too much about what's the intuition behind this or is there a good way to interpret this. What motivates these d variables?

---

> ### Author Response · Authors · 2017-12-23
> **Responses to Reviewer2**
>
> Thanks for your review and kind comments. In order to make the motivations and explanations to syntactic distance clearer, Section 4.2 has been rewritten accordingly to include the points we’ve mentioned here.
>
> The syntactic distance (d value) is motivated by trying to learn a scalar which indicates how semantically close each pair of words is. Our basic hypothesis is that words in the same constituent should have closer syntactic relation within themselves, and the syntactical proximity can be represented by a scalar value. From the tree structure point of view, the distance can be interpreted as positively correlated with the shortest path in the tree (in terms of the number of edges) between the two words. Syntactically the closer the two words are, the shorter this distance will be. Further, with the proof in Appendix C, we proved that by just using this scalar distance, a valid tree can be inferred.
>
> Mathematically the syntactic distance can also be naturally introduced from the stick breaking process, as a parametrization of \alpha in Eq. 6.
>
> From the viewpoint of computational linguistics, we did an extensive search and found some related work which tries to identify the beginning and ending words by just using local information, for example, Roark & Hollingshead, (2008). We have cited this work in the updated version.
>
> Thanks again for your kind review!

---

### Comment · AnonReviewer3 · 2017-11-18
**questions / comment**

The paper proposes a very cute idea. My questions/comment:

1. can you compute p(x_t+1|x0...x_t) (eq 3) by marginalising over g?

2. is the idea of using "syntactic distance" related to any linguistic theory?

3. I think eq 5 has a typo: is it g_i or g_t'?

4. the last line on page 4: d_{K-1} (capital K). Shouldn't d index start from 1? (you say that there are K-1 variables)

5. Eq 11: I think d doesn't need to be in [0, 1]. Did you try other activation functions (e.g Tanh)?

6. The line right after Eq 12: shouldn't t+1 be superscript? (It's better to be coherent with the notations above)

7. In 6.3, did you try any word embedding? As suggested by [1], word embeddings can be very helpful.


1. Le & Zuidema. Unsupervised Dependency Parsing: Let’s Use Supervised Parsers

---

> ### Author Response · Authors · 2017-11-21
> **Responses to questions / comment**
>
> Thank you for enlightening comments.
>
> Regarding "marginalizing over g":
> As discussed in section 4.1 and Appendix B, we replace discrete g by its expectation. Thus, we can have a computationally less expensive approximation for p(x_t+1|x0...x_t).
>
> Regarding "linguistic theory for 'syntactic distance'"
> The idea of using a "syntactic distance" is inspired by the binary parse tree, which is related to linguistic theory. We introduced the "syntactic distance" while trying to render the binary parse tree into a learnable, soft tree structure in the framework of language modeling. So it can be deemed as a set of boundaries which defines the binary parse tree.
>
> Regarding "try other activation functions"
> Thank you for this enlightening comment. We recently tried to replace sigmoid by ReLU, which makes the model achieve more stable performance regardless of different temperature parameter \tau.
>
> Regarding "try any word embedding"
> In this experiment, we want to prove the model's ability to learn from scratch, but pretrained word embedding can contain syntactic information. We will use word embedding in future work that focuses on obtaining better syntactic distance.
>
> This article will be further revised and polished according to your suggestions.

---

### Public Comment · (anonymous) · 2017-12-20
**Gain from explicitly modeling the syntactic structure**

This is really an interesting paper, which models the syntactic structure in a clever way. In the implementation (sec 5.1), an LSTMN is used to perform the recurrent update, where the syntactic gates g_i^t are used to modulate the content-based attention. So, I was wondering how much the model actually benefits from the syntactic modulation. Specifically, what the performance will be like without the syntactic modulation, i.e., with a standard LSTMN.

P.S. I've checked the original LSTMN paper, but the experiment setting (network size, hyper-parameters etc.) is different there.

---

> ### Author Response · Authors · 2017-12-23
> **An ablation test is added in section 6.2**
>
> Thanks for the comments and suggestions. In the new revision, we add the ablation test on PTB dataset. The "-Parsing Net" model in ablation test shows what the performance will be like without the syntactic modulation.

---

### Public Comment · ~Samuel_R._Bowman1 · 2018-02-25
**Public Code/Results?**

Neat work! Would you be willing to share (publicly or privately) your code and generated trees?

---

> ### Author Response · Authors · 2018-03-02
> **Code for language model experiment is released**
>
> The pytorch code for language model experiment is released at https://github.com/yikangshen/PRPN

---

### Author Response · Authors · 2018-03-02
**Code for language model experiment**

The pytorch code for language model experiment is released at https://github.com/yikangshen/PRPN

---

### Public Comment · ~Kyunghyun_Cho1 · 2018-04-05
**not fully unsupervised parsing**

thanks to the authors for promptly releasing the code public!

one thing i noticed from the released code is that early stopping for the experiments with sentence-level language modelling on penn treebank is done based on the F-1 score using the gold standard annotations:

https://github.com/yikangshen/PRPN/blob/master/main_UP.py#L228-L237

i believe this makes it less of unsupervised learning, and i couldn't find this setup mentioned in the paper. i kindly ask the authors to reflect this in the text to avoid misleading any reader.

---

> ### Author Response · Authors · 2018-04-10
> **Early stopping is not necessary**
>
> Thanks for pointing this out!
> We reran the experiment and found out that the early stopping by monitoring F1 is not necessary for unsupervised parsing task. As training loss decrease, the F1 almost always increase. Thus, we updated the code to use training loss as learning rate schedule and early stopping criteria:
> https://github.com/yikangshen/PRPN/blob/a1a8431499918a99f4689dca9428018ca2e256d9/main_UP.py#L229-L242
>
> Hope this can answer your question.

---

> > ### Public Comment · ~Yoon_Kim1 · 2018-08-07
> > **another question about F1 scores**
> >
> > Thanks very much for an interesting paper and also releasing the code!
> >
> > I was looking at the evaluation script for F1 (test function in this script: https://github.com/yikangshen/PRPN/blob/master/test_phrase_grammar.py) and found that it is calculated by first obtaining F1 for each sentence and then averaging across the corpus. My understanding, based on the evalb script (https://nlp.cs.nyu.edu/evalb/), is that F1 should be calculated at the corpus level (i.e. aggregating the true positives/false positives/false negatives at the corpus level).
> >
> > I was wondering if the results in the paper are based on sentence-level F1, and if so, how much they would change if the evaluation is based on corpus-level F1?
> >
> > Thanks!

---

### Public Comment · ~Phu-Mon_Htut1 · 2018-11-20
**In-depth analysis of the model**

We perform an in-depth analysis of the model to understand what it is learning and to compare it with the existing models that attempt to learn latent syntactic structure with the supervision from a downstream task. Although we discover a few issues with the model, we find that this model is robust and the learned parses agree with the Penn Treebank grammar significantly higher than chance. Please refer to our paper for more details: [paper : https://arxiv.org/abs/1808.10000 ], [code: https://github.com/nyu-mll/PRPN-Analysis].

---

### Decision · Program_Chairs · 2018-01-29
**ICLR 2018 Conference Acceptance Decision**

**Decision:**

Accept (Poster)

**Comment:**

Nice language modeling paper with consistently high scores. The model structure is neat and the results are solid. Good ICLR-type paper with contributions mostly on the ML side and experiments on a (simple) NLP task.